# PriFU: Capturing Task-Relevant Information Without Adversarial Learning

### Xiuli Bi
Chongqing Key Laboratory of Image
Cognition, Key Laboratory of
Cyberspace Big Data Intelligent
Security, Ministry of Education
Chongqing University of Posts and
Telecommunications
Chongqing, China
bixl@cqupt.edu.cn

### Yang Hu
Chongqing Key Laboratory of Image
Cognition, Key Laboratory of
Cyberspace Big Data Intelligent
Security, Ministry of Education
Chongqing University of Posts and
Telecommunications
Chongqing, China
s210201038@stu.cqupt.edu.cn

### Bo Liu[*]
Chongqing Key Laboratory of Image
Cognition, Key Laboratory of
Cyberspace Big Data Intelligent
Security, Ministry of Education
Chongqing University of Posts and
Telecommunications
Chongqing, China
boliu@cqupt.edu.cn

### Weisheng Li
Chongqing Key Laboratory of Image
Cognition, Key Laboratory of
Cyberspace Big Data Intelligent
Security, Ministry of Education
Chongqing University of Posts and
Telecommunications
Chongqing, China
liws@cqupt.edu.cn

### Pamela Cosman
Department of Electrical and
Computer Engineering
University of California, San Diego
San Diego, United States
pcosman@ucsd.edu

### Bin Xiao
Chongqing Key Laboratory of Image
Cognition, Key Laboratory of
Cyberspace Big Data Intelligent
Security, Ministry of Education
Chongqing University of Posts and
Telecommunications
Chongqing, China
xiaobin@cqupt.edu.cn

## Abstract

As machine learning advances, machine learning as a service (MLaaS) in the cloud brings convenience to human lives but also privacy risks, as powerful neural networks used for generation, classification or other tasks can also become privacy snoopers. This motivates privacy preservation in the inference phase. Many approaches for preserving privacy in the inference phase introduce multi-objective functions, training models to remove specific private information from users' uploaded data. Although effective, these adversarial learning-based approaches suffer not only from convergence difficulties, but also from limited generalization beyond the specific privacy for which they are trained. To address these issues, we propose a method for privacy preservation in the inference phase by removing task-irrelevant information, which requires no knowledge of the privacy attacks nor introduction of adversarial learning. Specifically, we introduce a metric to distinguish task-irrelevant information from task-relevant information, and achieve more efficient metric estimation to remove task-irrelevant features. The experiments demonstrate the potential of our method in several tasks. Our code will be available at: https://github.com/iwhoyoung/PriFU.

[*]Corresponding author.

## CCS Concepts

• **Computing methodologies** → **Computer vision tasks**; *Computer vision representations.*

## Keywords

Representation learning, Non-adversarial learning, Privacy preservation

**ACM Reference Format:**
Xiuli Bi, Yang Hu, Bo Liu, Weisheng Li, Pamela Cosman, and Bin Xiao. 2024. PriFU: Capturing Task-Relevant Information Without Adversarial Learning. In *Proceedings of the 32nd ACM International Conference on Multimedia (MM '24), October 28-November 1, 2024, Melbourne, VIC, Australia.* ACM, New York, NY, USA, 9 pages. https://doi.org/10.1145/3664647.3681216

## 1 Introduction

In recent years, machine learning development has led to the emergence of Machine Learning as a Service (MLaaS), such as human-like chat services, image generation services, and predictive analytic services. These services bring convenience to users but also privacy risks, as powerful neural networks can also be privacy snoopers. For example, when users using a plant identification cloud service upload photos, location information may be exposed due to the information-rich image including environmental information. Services for visual tasks usually involve uploading overly rich information, resulting in privacy risks. Therefore, *privacy preservation in the inference phase* (see Definition 3.1) comes into the spotlight, preserving the private information of the input in the inference phase (i.e., at test time).

To preserve users' privacy of the uploaded data, some approaches operate directly to manipulate the image, such as blurring [28] and masking [3]. They obfuscate key information in input images.

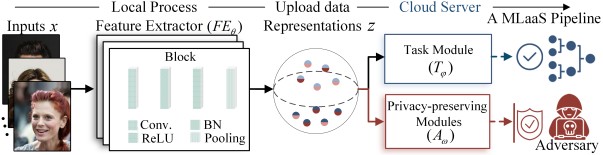

(a) The adversarial learning-based privacy-preserving framework

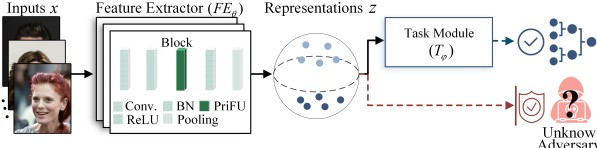

(b) Our proposed privacy-preserving framework

**Figure 1: Comparing our framework for preserving privacy in the inference phase with the previous framework. (a) There are additional modules for obfuscating specific private information (red). Privacy-preserving module complexity is proportional to privacy diversity. (b) Our proposed framework gets rid of adversarial learning to become more efficient, and may generalize beyond the specific target privacy.**

Some other approaches [5, 19, 21, 39] try to change the specific information for which users have privacy concerns. These methods can be considered as pre-processing before uploading images as MLaaS input. However, the pre-processed images may be out of the original distribution, leading to performance degradation in the original task. Thus this approach makes it difficult to achieve a good privacy utility trade-off. Meanwhile, image-level pre-processing has high costs in hardware and computational resources, which can be solved by model partitioning.

Model partitioning splits the model for services into two parts. One part is deployed on the user side for feature extraction, and the extracted representations will be uploaded to use the service instead of images. The second part offers the service by processing the uploaded representations. Although the black-box nature of neural networks makes such methods difficult to design, many novel and effective model partition methods for preserving privacy have been proposed. The methods [23, 26, 35, 40] fine-tune the representation $z$, including dimensionality reduction using Principal Component Analysis [1], adding elaborate noise, and reducing private information in the representation.

The above image pre-processing and representation fine-tuning approaches are two-stage, enabling privacy preservation independent from the MLaaS model. As mentioned above, this may result in an out-of-distribution image or representation, and thus degrade performance. To solve this issue, adversarial learning-based methods are intuitively introduced to achieve domain adaptation. Typically, the idea of generative adversarial networks (GANs) [6] is introduced, training the model for services when minimizing the user privacy risk. As shown in Fig. 1(a), privacy-preserving modules are added as the "adversaries" who try to steal private information. During training, a feature extractor is trained to prevent specific privacy leakage while ensuring good services (i.e., task). After training,

the feature extractor extracts privacy-preserving representations as updated data. Adversarial learning is effective for preserving privacy, and various adversarial learning-based methods were designed with different privacy-preserving modules [14–16, 36] or model training methods [2, 22, 41, 42].

Still, there are two issues with such methods: they are difficult to train, an inherent problem of adversarial learning and well known in GANs, and adversarial learning-based methods can only preserve the targeted privacy. Also, the privacy-preserving modules become more complex as more adversaries must be defended against. To overcome these issues, we propose a novel framework for privacy preservation in the inference phase by removing task-irrelevant information from the data, as shown in Fig. 1(b). Our framework can defend against diverse adversaries without knowledge of the adversaries. This work can be summarized as follows:

- We propose a framework to capture task-relevant information with only task-specific training, which can preserve privacy in the inference phase.
- We introduce gradient-based task relevance metrics, Average Relative Influence (AvgRI), to measure the influence of channel-level features on a task.
- Based on AvgRI, a model-agnostic plugin layer, dubbed the Privacy Filter Unit (PriFU), is proposed to weigh and filter channel-level features for preserving privacy.

## 2 Related Work

Preserving privacy in the inference phase, as a subfield of privacy preservation, is intended to protect the privacy of input data when using MLaaS in the cloud. To block risk exposure and keep effective services, there are two objectives: privacy and utility. The privacy goal is to keep as little private information as possible in the uploaded data, while the utility goal is to retain as many task-relevant features as possible in the uploaded data. Tremendous progress has been made in different ways to achieve the privacy and utility goals simultaneously.

**Image Pre-processing.** PrivacyNet generates privacy-preserving images [21] by a generator and multiple discriminators. One discriminator prompts the generator to improve the fidelity of the generated images, while the others for the privacy goal help the generator transform specific attributes. In [38], a bidirectional attribute classifier is proposed to transform specific attributes more finely.

**Representation Fine-tuning.** Samragh et al. [26] process pretrained representations by SVD and then remove the corresponding feature vectors in the order of smallest to largest singular values until the privacy goal is satisfied. Mireshghallah et al. [20] inject noise randomly sampled from the learned Laplace distribution to the representations in the inference phase.

**Adversarial learning.** As can be seen in Fig. 1 (a), adversarial learning-based methods introduce modules that act as the specific adversaries (privacy-preserving modules), prompting the feature extractor to learn to extract privacy-preserving representations during training. The methods [25, 43] employ a classifier as the adversary to infer specific privacy attributes. The classifier learns

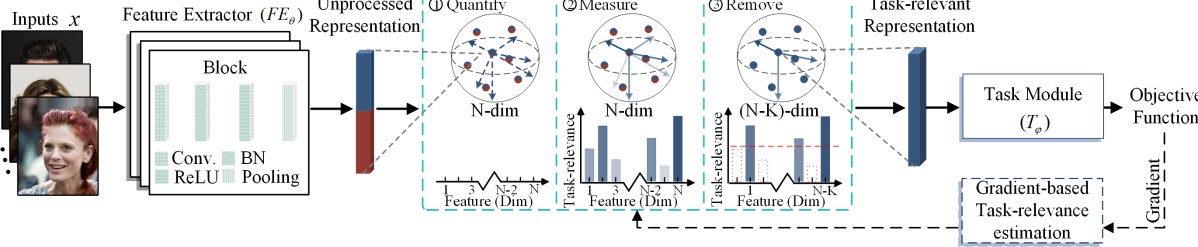

**Figure 2: Our proposed framework for preserving privacy in the inference phase. The arrows in the hypersphere represent different feature dimensions in the representation space. The color shade of the arrow indicates the task-relevance of features.**

attribute inference by minimizing the gap between prediction results and privacy labels. Among the methods, Controllable Invariance (CI) [43] trains the feature extractor to prevent attribute leaks by using the negative loss function of the adversary, while Maximum Entropy Adversarial Representation Learning (MaxEnt-ARL) [25] maximizes the information entropy of the private information. RAN [16] takes a generative network as the adversary (i.e., privacy-preserving modules in Fig. 1 (a)) rather than classifiers to train the feature extractor to prevent input image reconstruction. CPGAN [36] also takes a generative network as the adversary, but employs multiple alternative training strategies, such as Linear Ridge Regression [36], to pick the best one. PAN [15] takes a classification network and a generative network as the privacy-preserving modules. CFB [24] proposes a new variational approach to learn private representations. NoPeek-Infer [37] prevents reconstruction attacks by minimizing distance correlation between sensitive data. DISCO [30] learns a dynamic and data-driven pruning filter to selectively obfuscate sensitive information in the feature space.

Compared with these approaches, our framework aims to drop the task-irrelevant features, thus achieving privacy preservation in the inference phase, namely preserving privacy without predefining privacy. The simple training for tasks contributes to model convergence as no privacy-preserving modules are required.

## 3 Methodology

In this work, we focus on removing task-irrelevant information from the input data towards achieving *privacy preservation in the inference phase* (Definition 3.1). Our proposed framework can extend beyond the specific privacy preservation, which generalizes the concept of privacy preservation in the inference phase.

*Definition 3.1.* (Privacy Preservation in the Inference Phase) We consider an arbitrary data $\{x_u, y_u^{pri}\}$, where $x_u$ is the input data of user $u$, and $y_u^{pri}$ represents the ground truth of the private information. For any function $g : z_u \rightarrow \hat{y}_u^{pri}$, *Privacy preservation in the Inference Phase* trains a model $f : x_u \rightarrow z_u$ to prevent the function $g : z_u \rightarrow \hat{y}_u^{pri}$ from inferring the private information $y_u^{pri}$.

### 3.1 Threat Model and Target Model

**Threat Model:** We consider a threat model with full knowledge of the feature extractor and task module ($FE_\theta$ and $T_\phi$), the training dataset ($D_{train}$) and privacy-preserving data (i.e., representation $z$ shown in Fig. 1), since the user modules are usually downloaded

---

**Algorithm 1** General Attack Algorithm

1: **Input:** Annotated training datasets $D_{train}$, trained feature extractor $FE_\theta$, uploaded representation $z_u$.
2: **Output:** Predicted private information $\hat{y}_{pri}$.
3: Query $FE_\theta$ with $D_{train}$ and collect $\{(FE_\theta(x_i), y_i)|(x_i, y_i) \in D_{train}\}$.
4: Based on $\{(FE_\theta(x_i), y_i)|(x_i, y_i) \in D_{train}\}$, train the attack model $g_\phi : FE_\theta(x_i) \rightarrow y_i$.
5: Predict $\hat{y}_u^{pri} = g_\phi(z_u)$.

---

from the cloud server and the training dataset is usually public in the community. Hence, in the inference phase, the threat model tries to infer private information from the privacy-preserving data, as shown in Algorithm 1. The threat model trains an attack model $g_\phi$ to approximate the intended private data $\hat{y}_u^{pri}$. In previous work, reconstruction attacks and sensitive attribute inference attacks [31] are widely used to evaluate the capacity of privacy-preserving methods. The reconstruction attack trains a generative model to reconstruct the input data with the labeled training dataset, while the sensitive attribute inference attack trains a classifier to classify the sensitive attribute with the labeled training dataset.

Although the attack pipeline is usually as shown in Algorithm 1, the exact attack model $g_\phi$ implemented is unknown as well as the intended private information of the uploaded data. Thus the target model for preserving privacy is desired to preserve diverse private information. This inspires the intuition that the target model should only retain the task-relevant information, rather than obfuscating the specific private information against the exact attack. Further, the target model is given here.

**Target Model:** For any user data $x_u$ and any attack model $g_\phi$, the feature extractor ($FE_\theta$) removes private information from the user data $x_u$, and outputs the privacy-preserving representation $z$. The attack model $g_\phi$ can not get the private information by processing the representation $z$, while the task module $T_\phi$ can offer the service.

### 3.2 Proposed Framework to Capture Task-Relevant Information

As mentioned above, we try to remove task-irrelevant information in the representation. Then, a framework is proposed as shown in Fig. 2. The framework proceeds to:

(1) quantify the information and divide it into units;

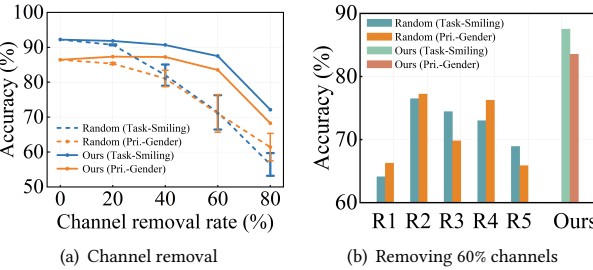

(a) Channel removal     (b) Removing 60% channels

**Figure 3: Task performance vs. privacy performance of the ResNet18s trained on CelebA dataset after removing some channels. (a): Comparing random channel removal with channel filtering based on the AvgRI (Ours) when setting different channel removal rates. (b): Comparing random channel removal (R1-R5) with channel filtering based on the AvgRI when some channels are removed.**

(2) measure the task-relevance of each unit of information;

(3) remove the units of information with task-relevance below a preset threshold.

As shown in Fig. 2, the uploaded information is represented by N-dim features in the N-dim representation space. We measure the task-relevance of each dimension in some way, and transform the N-dim representation space into the (N-K)-dim representation space by removing the K dimensions which have task-relevance below a preset threshold. Therefore, when an input data is projected into the (N-K)-dim representation space as the uploaded data, the uploaded data contains almost no private information. This enables privacy preservation in the inference phase.

To propose an effective framework, we first explore how to quantify the input data information. A straightforward solution is to divide the information into units based on the channel partitioning, as neural networks use a multi-channel architecture to capture different features in parallel. To assess this solution, we train different models for classification, and randomly remove some of the channels of the layer that are closest to the classifier. Also, another classifier is trained to classify the privacy as an evaluation of the privacy-preserving performance. As shown in Fig. 3(a), some of the results suggest that the retained information contains more task-relevant features and fewer task-irrelevant features. Therefore, we next find a metric that can measure the task-relevance of each feature channel, and filter the channels by comparing the metric rather than removing them randomly.

### 3.3 Gradient-Based Task-Relevance Metrics

Many white-box attack methods [7, 13, 18] in the adversarial attack field drastically perturb model results by changing the pixels which have large gradient w.r.t the loss, since the magnitude of the gradient can indicate the influence of the pixel on the model result. However, the influence is typically noisy due to the noisy gradients [4, 10, 33], and some weakly supervised object localization methods [32, 34] mitigate this issue. Hence, inspired by such white-box attacks and weakly supervised object localization, we first introduce a quantitative metric, Relative Influence (RI), to assess the influence

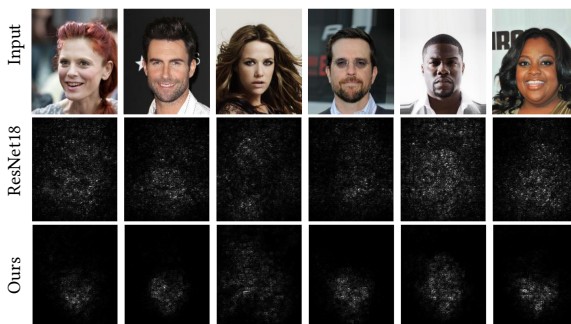

**Figure 4: Visualizations of normalized RIs of pixels on loss. Both our proposed privacy-preserving framework (Fig. 1 (b)) and ResNet18 are trained on CelebA [17].**

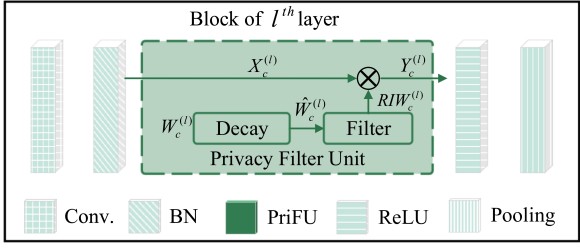

**Figure 5: An illustration of the proposed Privacy Filter Unit in the block of the $l^{th}$ layer. $W_c^{(l)}$ is a learnable parameter.**

of the input on the output. We get the RI by calculating the absolute gradient of the input w.r.t the output. Given an input $x$ and a scalar function $M$, the corresponding RI can be calculated by

$$RI^M(x) = \left| \frac{\partial M(x)}{\partial x} \right|. \tag{1}$$

Fig. 4 shows several examples where input $x$ is a pixel and the scalar function $M$ is ResNet18 [8] with cross-entropy loss.

To reduce the noise disturbance in Eq. (1) due to noisy gradients [4, 10, 33], Average Relative Influence (AvgRI) is introduced. Given $N$ inputs $\{x_n\}_{n=1}^N$ sampled from a data distribution $p_{data}$ and a scalar function $M$, the AvgRI of the data $x$ on the output is:

$$AvgRI^M(x) = E_{x \sim p_{data}}\left[ \left| \frac{\partial M(x)}{\partial x} \right| \right] \approx \frac{1}{N}\sum_{n=1}^{N} \left| \frac{\partial M(x_n)}{\partial x_n} \right|. \tag{2}$$

We approximate the AvgRI of the channel-level features on loss by calculating the average of the RI of $N$ sampled data to measure the task-relevance of the channel-level features.

### 3.4 Gradient-Based Task-Relevance Estimation

After measuring the task relevance of channel-level features in different layers, we rank the task relevance and remove channels with low task relevance so that our framework seems to be achieved. However, structural changes caused by channel removal block this solution, since the calculation of AvgRI is based on the specific function $M$ as shown in Eq. (2). If channel removal leads to model

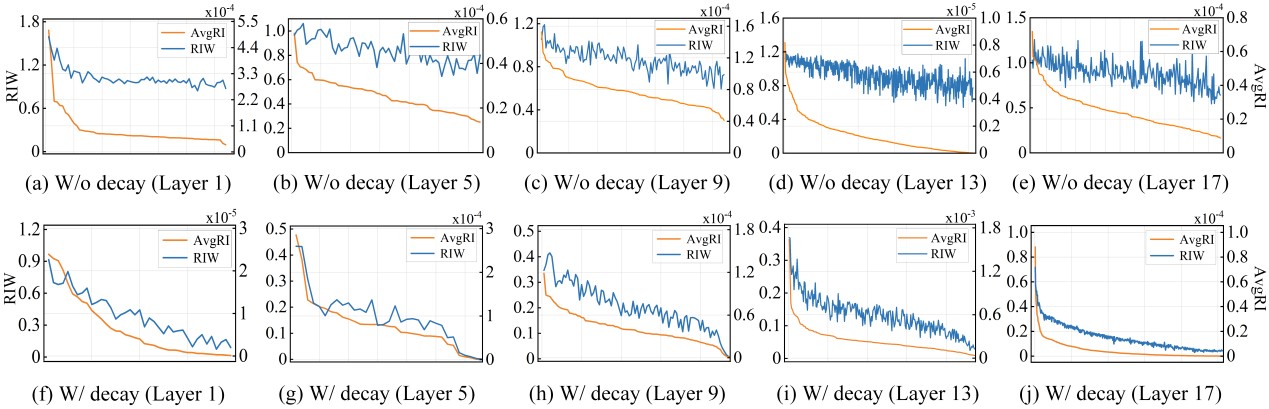

(a) W/o decay (Layer 1)  (b) W/o decay (Layer 5)  (c) W/o decay (Layer 9)  (d) W/o decay (Layer 13)  (e) W/o decay (Layer 17)

(f) W/ decay (Layer 1)  (g) W/ decay (Layer 5)  (h) W/ decay (Layer 9)  (i) W/ decay (Layer 13)  (j) W/ decay (Layer 17)

**Figure 6: The charts of $W_c^{(l)}$ and $AvgRI^{M_c^{(l)}}(X_c^{(l)})$ in different channels $c$. We rank the channels by $AvgRI^{M_c^{(l)}}(X_c^{(l)})$ and show the corresponding $W_c^{(l)}$. The results are from ResNet18 with and without the decay component.**

retraining and AvgRI recalculation, the increasing time cost is intolerable due to the calculation of the AvgRI over all the training data. To solve this issue, we propose a plugin layer, PriFU, to estimate the task relevance of features ($X_c^{(l)}$ in Eq. (3)) since we can get the gradients in back-propagation, and then PriFU filters the features by comparing the estimated task relevance. PriFU is shown in Fig. 5 and can be written as

$$Y_c^{(l)} = RIW_c^{(l)} \cdot X_c^{(l)}, \tag{3}$$

where $X_c^{(l)}$ represents the feature in channel $c$ of the $l^{th}$ block and $Y_c^{(l)}$ is the counterpart after filtering by PriFU. The $RIW_c^{(l)}$ is used to filter the features by thresholding the $\widehat{W}_c^{(l)}$ of the channel-level feature $X_c^{(l)}$. Ideally, the $\widehat{W}_c^{(l)}$ is proportional to the AvgRI. We will remove the feature $X_c^{(l)}$ if the $\widehat{W}_c^{(l)}$ of the channel-level feature $X_c^{(l)}$ is below the threshold, which can be written as

$$RIW_c^{(l)} = \begin{cases} \widehat{W}_c^{(l)} & \widehat{W}_c^{(l)} > \alpha \cdot Max(\widehat{W}_c^{(l)}) \\ 0 & \widehat{W}_c^{(l)} \leq \alpha \cdot Max(\widehat{W}_c^{(l)}) \end{cases}, \tag{4}$$

where $\alpha$ is the hyper-parameter to change the utility, and $Max(\widehat{W}_c^{(l)})$ represents the highest value among the task relevance estimates $\widehat{W}_c^{(l)}$ of features in all channels, which can help to adaptively deal with numerical differences in different layers. To better replace the role of AvgRI, we introduce a decay component as follows:

$$\widehat{W}_c^{(l)} = W_c^{(l)} \tag{5}$$

in forward propagation and

$$\nabla \widehat{W}_c^{(l)} = \frac{\partial \ell}{\partial W_c^{(l)}} + \beta \cdot W_c^{(l)} \tag{6}$$

in backward propagation, where $\beta$ is the hyper-parameter to tune the decay rate, and $\frac{\partial \ell}{\partial W_c^{(l)}}$ is the gradient of $W_c^{(l)}$ w.r.t. the loss. The modified gradient $\nabla \widehat{W}_c^{(l)}$ is used to calculate the updated $W_c^{(l)}$ instead of the original gradient.

We now clarify theoretically that the decay component helps $\widehat{W}_c^{(l)}$ to replace the role of $AvgRI^{M_c^{(l)}}(X_c^{(l)})$. Based on Eq. (2), the

AvgRI of $X_c^{(l)}$ can be written as

$$AvgRI^{M_c^{(l)}}(X_c^{(l)}) = E\left[\left|\frac{\partial \ell}{\partial X_c^{(l)}}\right|\right] = E\left[\left|RIW_c^{(l)} \cdot \frac{\partial \ell}{\partial Y_c^{(l)}}\right|\right], \tag{7}$$

where $M_c^{(l)}$ is the rest after the Batch Normalization (BN) layer in the $l^{th}$ block of the model with loss function, and $\ell$ is the loss. If $RIW_c^{(l)}$ is greater than 0, $AvgRI^{M_c^{(l)}}(X_c^{(l)})$ can be written as

$$AvgRI^{M_c^{(l)}}(X_c^{(l)}) = RIW_c^{(l)} \cdot E\left[\left|\frac{\partial \ell}{\partial Y_c^{(l)}}\right|\right] \; s.t. \; RIW_c^{(l)} > 0. \tag{8}$$

Assuming that all the $RIW_c^{(l)}$ are greater than 0, the ratio of the AvgRI of two channel-level feature maps output by the BN layer in the $l^{th}$ block on the loss is

$$\frac{AvgRI^{M_{c1}^{(l)}}(X_{c1}^{(l)})}{AvgRI^{M_{c2}^{(l)}}(X_{c2}^{(l)})} = \frac{RIW_{c1}^{(l)}}{RIW_{c2}^{(l)}} \cdot \frac{E\left[\left|\frac{\partial \ell}{\partial Y_{c1}^{(l)}}\right|\right]}{E\left[\left|\frac{\partial \ell}{\partial Y_{c2}^{(l)}}\right|\right]}, \tag{9}$$

where $c1$ and $c2$ represent two different channels. Assuming the difference between $RIW_c^{(l)}$ of different channels is much larger than the difference between the $E\left[\left|\frac{\partial \ell}{\partial Y_c^{(l)}}\right|\right]$, the following relationship holds:

$$RIW_c^{(l)} \propto AvgRI^{M_c^{(l)}}(X_c^{(l)}), \tag{10}$$

and we can filter out the channel-level features when $RIW_c^{(l)}$ is less than the threshold. To satisfy the assumptions of Eq. (10), we introduce the decay component to increase the difference between $RIW_c^{(l)}$ of different channels. This improves the task-relevance estimation, which is supported by Fig. 6. Also, Fig. 6 demonstrates that $AvgRI^{M_c^{(l)}}(X_c^{(l)})$ and $W_c^{(l)}$ are overall positively correlated.

Given an example based on Stochastic Gradient Descent (SGD), the updated $W_c^{(l)}$ after one iteration is 0 if $\widehat{W}_c^{(l)} \leq \alpha \cdot Max(\widehat{W}_c^{(l)})$ (Eq. (4)), or otherwise

$$W_c^{(l)} - lr \cdot \nabla \widehat{W}_c^{(l)} = (1 - lr \cdot \beta)W_c^{(l)} - lr \cdot \frac{\partial \ell}{\partial W_c^{(l)}}, \tag{11}$$

**Table 1: Details of the datasets. Privacy 1 is the target privacy attribute; Privacy 2 is one of the unprotected attributes if the methods only preserve the specific privacy.**

| Dataset | CelebA | LFW | Cifar10 |
|---|---|---|---|
| Type | facial | facial | nature |
| Size | 218×178 | 250×250 | 32×32 |
| Training | 62,770 | 9,635 | 50,000 |
| Testing | 7,227 | 3,510 | 10,000 |
| Task | smiling | smiling | living |
| Privacy 1 (Pri. 1) | gender | pale skin | 10-class |
| Privacy 2 (Pri. 2) | young | young | – |

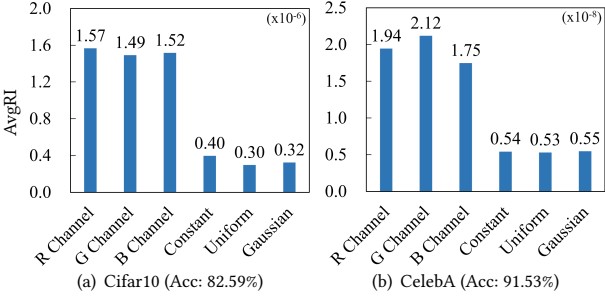

(a) Cifar10 (Acc: 82.59%)  (b) CelebA (Acc: 91.53%)

**Figure 7: AvgRI of different channels of the 6-channel inputs on the loss. *R*, *G* and *B* represent the three channels of RGB images. *Constant*, *Uniform* and *Gaussian* represent the introduced noise channels of the same size as the input. *Constant*: all pixel values are 1. *Uniform*: The pixel values are sampled from a uniform distribution of 0 to 1. *Gaussian*: The pixel values are sampled from a Gaussian distribution with mean 0.5 and variance 1.**

where $lr$ is the learning rate and $\nabla\widehat{W}_c^{(l)}$ is calculated by Eq. (6). As shown in Eq. (11), the decay component with a suitable hyperparameter $\beta$ can avoid large initial values of $W_c^{(l)}$ leading to small differences among the values of $\widehat{W}_c^{(l)}$ of different channels in the $l^{th}$ block. The decay component does not lead to any value of $W^{(l)}$ less than 0 while ensuring that $lr \cdot \beta < 1$.

## 4 Experiments

In this section, we show the effectiveness of AvgRI, and introduce experiments on the hyperparameters $\alpha$ and $\beta$. We also demonstrate the model-agnostic nature of PriFU by inserting it into neural networks of different depths and structures. An ablation study is also introduced. Finally, we compare our method with others.

### 4.1 Experimental Setup

**Datasets.** We use three public datasets to evaluate privacy preservation performance in the inference phase. Table 1 shows these datasets in detail. The datasets have been widely used in previous work [15, 16, 24, 25, 30, 37, 43]. CelebA [17] is a dataset of celebrity images labeled with 40 binary facial attributes, while LFW [12] consists of face images labeled with 40 binary facial attributes. For the selection of facial attributes in the experiments, we follow the widely used setups shown in Table 1. Although human-related

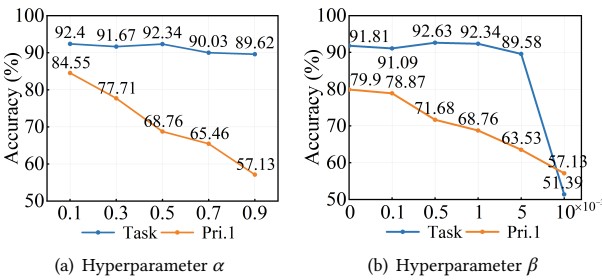

(a) Hyperparameter $\alpha$  (b) Hyperparameter $\beta$

**Figure 8: The results of our framework with different hyperparameters $\alpha$ and $\beta$ trained on CelebA Dataset.**

privacy issues are of concern, the privacy preservation of natural images should not be ignored. Therefore, we conducted experiments on Cifar10 [11], a single-label dataset which consists of 10-class nature images. Following the evaluation presented in [25], we divided the classes into two supersets, living ('bird', 'cat', 'deer', 'dog', 'frog', 'horse') and non-living ('airplane', 'automobile', 'ship', 'truck') objects, as the label for the task. The classification of the 10 basic classes is then used as the privacy-preserving task. This dataset is introduced to show the ability to deal with supersets and subsets.

**Evaluation Setup.** We intend to evaluate the information contained in the representations. A classifier is used to capture input data attribute information by classifying the attributes, while a decoder is used to reconstruct the entire images. Accuracy is introduced to measure the amount of the specific attribute information in the representation, while Peak Signal to Noise Ratio (PSNR) and Structural Similarity (SSIM) are introduced to measure the amount of input data information remaining in the representation.

**Implementation Details.** In our framework, we insert the PriFUs following the normalization layers of each block and the default model is ResNet18. The data augmentation pipeline consists of random horizontal flipping and normalization. By default, we trained our model with a learning rate of 0.1 and optimized by SGD. The learning-based weights $W_c^{(l)}$ are initiated with 0.1. The hyperparameter $\beta$ is 0.001, while the hyperparameter $\alpha$ is tuned for trading off privacy preservation and tasks.

### 4.2 Evaluation Experiments

**Experiments on Effectiveness of AvgRI.** We introduce this experiment to support that AvgRI can reflect the influence of multidimensional inputs on the output. ResNet18 is trained with 6-channel inputs (RGB and three noise channels) and the calculated AvgRIs of the different input channels on the loss are shown in Fig. 7. The AvgRIs of the RGB channels are much higher than those of the noise channels, which align well with the noise effects. The models work normally. The accuracy of the 10-class task on Cifar10 is 82.59% (84.39% before adding noise), while the accuracy of the 2-class task (smiling/no-smiling) on CelebA is 91.53% (92.22% before adding noise).

**The Role of Hyperparameters.** We alternately fix one hyperparameter and show the effect of another in Fig. 8. Hyperparameter

**Table 2: The ablation study of PriFU against reconstruction attack by a generative model on CelebA. Baseline represents the trained ResNet18.**

| Decay | Filter | Setup | Task↑ | PSNR↓ | SSIM↓ |
|-------|--------|-------|-------|-------|-------|
| ✗ | ✗ | Baseline | 92.22 | 15.61 | 0.6168 |
| ✓ | ✗ | Setup 1 | 92.18 | 15.44 | 0.6082 |
| ✗ | ✓ | Setup 2 | 88.53 | 12.17 | 0.5205 |
| ✓ | ✓ | Ours | 89.62 | 11.33 | 0.5078 |

**Table 3: Performance of our framework against reconstruction by a generative model on CelebA.**

| Backbone | Task↑ Baseline | Task↑ Ours | PSNR↓ Baseline | PSNR↓ Ours | SSIM↓ Baseline | SSIM↓ Ours |
|----------|---------|------|---------|------|---------|------|
| ResNet18 | 92.22 | 89.62 | 15.61 | 11.33 | 0.6168 | 0.5078 |
| ResNet152 | 91.93 | 88.79 | 15.49 | 11.44 | 0.6152 | 0.5090 |
| VGG16 | 92.18 | 87.45 | 18.03 | 11.38 | 0.6889 | 0.5100 |
| DenseNet121 | 92.06 | 88.67 | 16.88 | 11.32 | 0.6467 | 0.5080 |

$\alpha$ can be used to alter the privacy-utility tradeoff, while hyperparameter $\beta$ is used to change the fixed point in the solution space. **Ablation Study.** Ablation of the proposed PriFU is shown in Table 2 and Fig. 9(a). The filter component is the key of PriFU and the decay component further improves PriFU, which empirically verifies the theory of Section 3. Concretely, classification results for the *young* attribute (Pri. 2) and the image reconstruction results show a significant improvement with the help of the decay component.

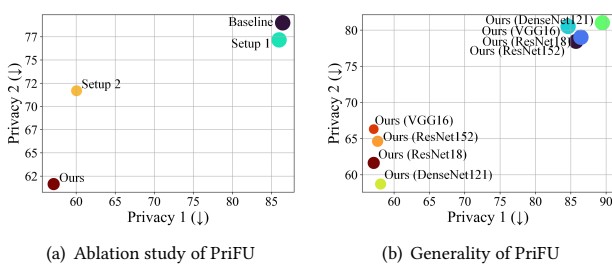

(a) Ablation study of PriFU   (b) Generality of PriFU

**Figure 9: Privacy-preserving performance of our proposed PriFU against attribute inference attack with different backbones on CelebA. Bubble size indicates task performance. Privacy (1/2) are described in Table 1.**

**Generality of PriFU.** We show the privacy-preserving performance of our framework with different backbones in Table 3 and Fig. 9(b); PriFU works well when employing backbone networks [8, 9, 29] of different depths and structures. Our framework with a deep backbone network (e.g., ResNet152), demonstrates that PriFU does not cause vanishing gradients, due to the role of the normalization layer in stabilizing the gradient [27]. Baseline is the model trained only for the task.

### 4.3 Comparative Experiments

Among the previous privacy-preserving tasks [15, 16, 24, 25, 30, 37, 43], the attribute inference attack and reconstruction attack are

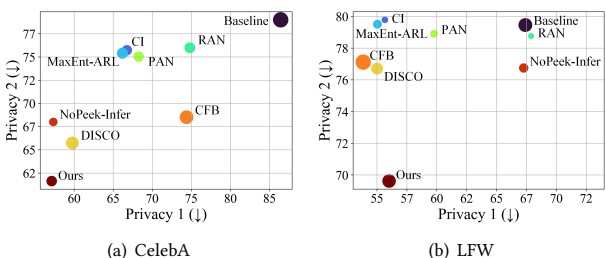

(a) CelebA   (b) LFW

**Figure 10: Privacy-preserving performance of methods against attribute inference attack on CelebA and LFW. Bubble size indicates task performance.**

**Table 4: Privacy-preserving performance of different methods against reconstruction attack on CelebA and LFW. Baseline represents the ResNet18 trained only for the task.**

| METHOD | CELEBA TASK↑ | CELEBA PSNR↓ | CELEBA SSIM↓ | LFW TASK↑ | LFW PSNR↓ | LFW SSIM↓ |
|--------|------|------|------|------|------|------|
| BASELINE | 92.22 | 15.61 | 0.6168 | 90.05 | 14.32 | 0.5973 |
| CI | 89.41 | 15.02 | 0.5923 | 84.65 | 14.56 | 0.6077 |
| MAXENT-ARL | 89.92 | 15.48 | 0.6025 | 86.49 | 14.23 | 0.5807 |
| RAN | 89.73 | 13.37 | 0.5535 | 84.11 | 12.05 | 0.5342 |
| PAN | 89.51 | 12.88 | 0.5393 | 85.58 | 12.32 | 0.5253 |
| DISCO | 90.91 | 11.59 | 0.4675 | 88.63 | 12.12 | 0.5319 |
| CFB | 91.35 | 12.08 | 0.5260 | 91.05 | 12.88 | 0.5482 |
| NOPEEK-INFER | 88.36 | 14.70 | 0.5531 | 86.89 | 14.06 | 0.5950 |
| OURS | 89.62 | 11.33 | 0.5078 | 89.85 | 12.40 | 0.5354 |

**Table 5: Privacy-preserving performance of methods on Cifar10. Baseline is the ResNet18 trained only for the task.**

| Method | Task↑ | Attr. infer. Privacy 1↓ | Reconstruction PSNR↓ | Reconstruction SSIM↓ |
|--------|-------|-------------|-------|-------|
| Baseline | 94.13 | 51.39 | 14.81 | 0.3816 |
| CI | 92.05 | 27.40 | 14.11 | 0.3099 |
| MaxEnt-ARL | 88.87 | 24.18 | 14.02 | 0.3043 |
| RAN | 88.41 | 34.36 | 12.97 | 0.2552 |
| PAN | 91.45 | 25.18 | 12.89 | 0.2690 |
| DISCO | 91.35 | 24.87 | 12.27 | 0.2315 |
| CFB | 88.61 | 31.12 | 12.25 | 0.2313 |
| NoPeek-Infer | 86.60 | 33.53 | 13.84 | 0.3281 |
| Ours | 90.21 | 21.83 | 12.41 | 0.2385 |

commonly used to evaluate the effectiveness of privacy-preserving methods. We also introduce them to demonstrate the superiority of our framework on the unknown privacy-preserving tasks, simply by training for the task. Also, our privacy-preserving framework yields comparable results on the privacy-preserving tasks targeted by the methods for active defense.

**Privacy Preservation for Unknown Privacy.** Fig. 10 shows that these existing privacy-preserving methods for active defense are difficult to generalize to unknown adversaries. For our proposed privacy-preserving framework, although all the adversaries are unknown, our framework shows comparable results against the

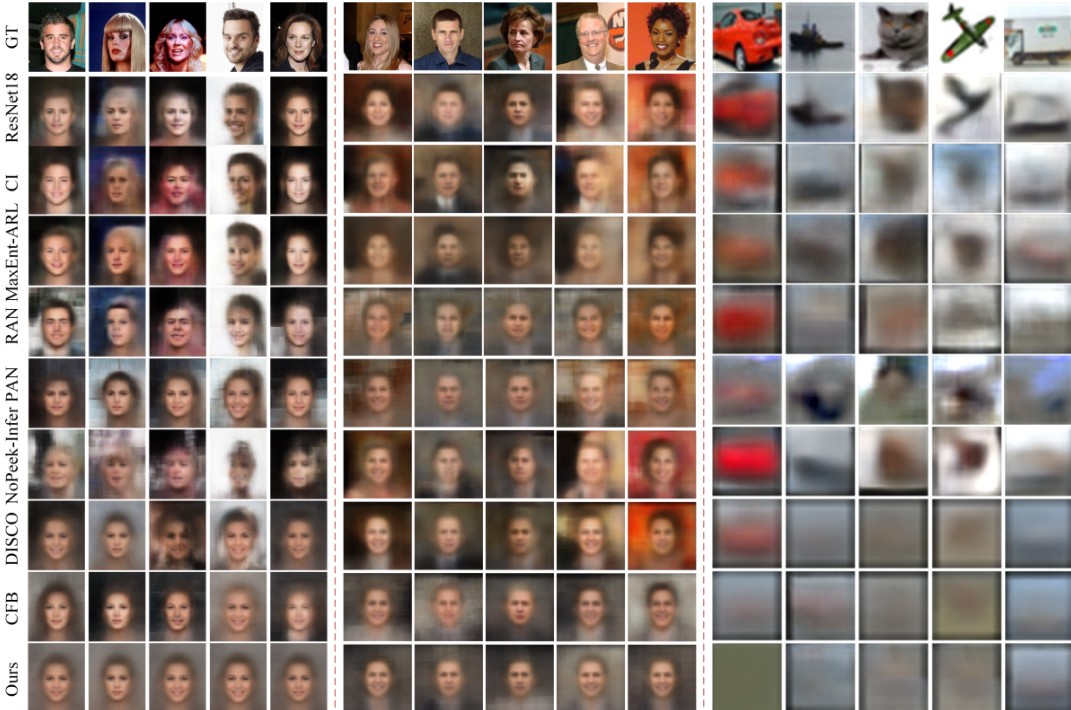

**Figure 11: Examples of reconstructed images on three datasets. GT means ground truth.**

adversaries targeted by the active defense methods and otherwise achieves SOTA results. This is our main advantage.

**Comparing with Targeted Privacy Preservation.** Attribute inference attack and reconstruction attack are two benchmark tasks for privacy preservation in the inference phase. We extend the experiments to three datasets to show the adaptability of our privacy-preserving framework to diverse data. CI [43] and MaxEnt-ARL [25] are specific privacy-preserving methods for attribute inference attack and are effective. However, Fig. 10 and Tables 4 and 5 illustrate neither of them is good against reconstruction attack. Also RAN [16] is effective only against reconstruction attack. The adversarial learning-based methods may only handle the targeted adversaries well.

## 5 Discussion of Limitations

The proposed Privacy Filter Units are introduced in different layers, filtering the features in different feature space and helping disentangling features. However, the feature entanglement can not be overcome completely. Therefore, we extend the evaluation to highly correlated attributes; the results are shown in Table 6. The results for the first case suggest that there is no feature entanglement between the action and color features of a same region, such as the features of smiling and wearing lipstick. This is probably because color is determined by multi-channels of input, and the information is divided into units through channel partitioning in this work. As for smiling and high-cheekbone which are coupled in one channel, the adversarial learning-based privacy-preserving methods do yield slightly better results for the specific attribute,

but their enhanced privacy presentation comes with sacrificing downstream task performance (Table 6, last two rows). This implies that these methods are also troubled by coupled features.

**Table 6: Classification of methods with different privacy-utility tradeoffs for highly correlated attributes on CelebA.**

| Attribute | ResNet18 | CI | MaxEnt-ARL | Ours |
|---|---|---|---|---|
| Task (Smiling↑) | 91.65 | 88.74 | 90.21 | 89.62 |
| Pri. (Wearing-lipstick↓) | 81.62 | 59.47 | 67.59 | 62.18 |
| Task (Smiling↑) | 91.65 | 89.47 | 90.56 | 89.62 |
| Pri. (High-cheekbone↓) | 84.93 | 80.75 | 81.45 | 83.15 |
| Task (Smiling↑) | 91.65 | 75.87 | 73.88 | 85.86 |
| Pri. (High-cheekbone↓) | 84.93 | 65.56 | 71.41 | 81.90 |

## 6 Conclusions

This paper shows a paradigm to capture task-relevant information: (1) quantifying the information as units; (2) measuring the task relevance of the information units; and (3) removing the information units with low task-relevance. Consequently, a framework is presented to capture task-relevant information with only task-specific training, which can be used to preserve the privacy of the uploaded data. To achieve this framework, we introduce a gradient-based task-relevance estimation to improve the effectiveness. Our proposed method converges easily due to the absence of adversarial learning. The removal of task-irrelevant information from user-uploaded data allows our method to generalize beyond a specific privacy attack in the inference phase.

# Acknowledgments

This work was supported in part by the National Natural Science Foundation of China under Grant 62172067 and Grant 62376046, the Natural Science Foundation of Chongqing for Distinguished Young Scholars under Grant CSTB2022NSCQ-JQX0001, in part by the Natural Science Foundation of Chongqing under Grant CSTB2023NSCQ-MSX0341, the Science and Technology Research Program of Chongqing Municipal Education Commission (Grant No. KJQN202200635), and in part by the Chongqing Graduate Student Research Innovation Project CYS23429.

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
