# OpenReview forum: "PriFU: Capturing Task-Relevant Information Without Adversarial Learning"
_acmmm.org/ACMMM/2024/Conference — MM2024 Poster_

### Official Review · Reviewer_UV6n · 2024-05-24

**Rating:** 4
**Confidence:** 3

**Summary:**

This work proposed a novel framework for privacy preservation in the inference phase of machine learning as a service (MLaaS). The proposed method, PriFU (Privacy Filter Unit), focuses on removing task-irrelevant information from the input data without the need for adversarial learning. The framework employs gradient-based task relevance metrics, specifically Average Relative Influence (AvgRI), to measure and filter channel-level features, thereby enhancing privacy while maintaining task performance.

**Strengths:**

(1)	This work introduces of the PriFU method, which captures task-relevant information without relying on adversarial learning and is an innovation in privacy-preserving machine learning. The use of gradient-based task relevance metrics (AvgRI) to quantify and filter task-relevant features is an effective strategy to enhance privacy preservation.
(2)	The paper conducts extensive experiments across multiple datasets (CelebA, LFW, Cifar10) and different neural network backbones (ResNet18, ResNet152, VGG16, DenseNet18). The results consistently demonstrate the effectiveness and generality of the PriFU method.
(3)	The methodology is well-structured and clearly described, with detailed explanations of the threat model, target model, and the proposed framework. Diagrams and figures effectively support the text, enhancing understanding.

**Limitations:**

（1）The Task-Relevance Estimation method is highly coupled with the data, and definitions of privacy and sensitive information vary across different datasets. Therefore, whether this method can be universally applied in real-world scenarios is a question that requires thorough analysis and validation
（2）Further analysis is needed on the relationship between the effectiveness of privacy information filtering and its impact on target model performance, particularly focusing on trade-offs and potential performance degradation in practical applications.

**Suitability:**

3

---

### Official Review · Reviewer_QM7j · 2024-05-25

**Rating:** 4
**Confidence:** 2

**Summary:**

This paper addresses privacy risks in machine learning as a service (MLaaS) by proposing a method for privacy preservation during the inference phase. Unlike adversarial learning-based approaches, which suffer from convergence difficulties and limited generalization, the proposed method removes task-irrelevant information without requiring knowledge of privacy attacks. It introduces a metric to distinguish task-relevant from task-irrelevant information and achieves efficient metric estimation to remove unnecessary features. Experimental results demonstrate the effectiveness of the method across various tasks.

**Strengths:**

1.	The topic is of great significance and the paper is well-written in general. The proposed method avoids adversarial learning, reducing complexity and improving convergence.
2.	The method balances privacy preservation and task performance, demonstrated through significant improvements in experiments across multiple datasets.
3.	The experiments are also comprehensive to some extent.

**Limitations:**

1.	The paper uses different channels to represent different features. However, for deep learning models, there are other ways to represent features. Please add ablation experiments to discuss the effectiveness of this approach, such as comparing PriFU with random channel drop or other feature representation methods.

2.	Increase the discussion on generalization. Using channels to represent features is suitable for image-related models, but in other models, such as NLP models, features are usually represented differently. The authors could add experiments to demonstrate the generalizability of the proposed method.

3.	A section discussing limitations should be added, such as the additional computational overhead introduced by PriFU.

4.	Reformat the labels in Figures 9 and 10 to avoid overlapping.

**Suitability:**

3

---

### Official Review · Reviewer_H1xD · 2024-05-27

**Rating:** 4
**Confidence:** 2

**Summary:**

The paper proposes a method aims to preserve privacy by removing task-irrelevant information without relying on adversarial learning. The method consists of the Average Relative Influence (AvgRI) metric to measure the influence of channel-level features on a task and a gradient-based task-relevance estimation.

**Strengths:**

- the paper includes a detailed theoretical justification for the proposed methods
- the paper conducts extensive experiments on public datasets to evaluate the effectiveness of the proposed method, it also compares the performance of the method with existing adversarial learning-based privacy-preserving methods, showing that the proposed method achieves state-of-the-art results.

**Limitations:**

- The paper acknowledges the difficulty in dealing with correlated features but does not provide a robust solution for the issue.

**Suitability:**

2

---

### Official Review · Reviewer_r5AD · 2024-05-30

**Rating:** 4
**Confidence:** 2

**Summary:**

This paper addresses the challenge of privacy preservation during the inference stage. Motivated by the convergence issues and generalization bottlenecks of adversarial learning-based strategies, this paper proposes a method that includes gradient-based task relevance metrics and a model-agnostic plugin layer to remove task-irrelevant information without requiring adversarial training. Experiments are conducted on the human face dataset CelebA and the natural image dataset CIFAR-10.

**Strengths:**

1. The task of privacy preservation during the inference phase is well-motivated by its broader impact on MLaaS users' privacy and the limitations of alternative strategies.
2. The methodology is intuitively motivated, as existing strategies rely on adversarial training, which suffers from poor convergence and generalizability. Introducing a method that bypasses the need for adversarial training is well-justified.
3. Conducting ablation studies and including a natural image dataset as part of the experiments is commendable.

**Limitations:**

1. The paper needs proofreading, as it contains several careless errors, including typos and an unmodified template in the CCS concepts section.
2. The experimental results do not appear promising, which undermines the practical significance of the proposed method.

**Suitability:**

2

---

### Meta-Review · Area_Chair_y9N1 · 2024-07-02

**Recommendation:** Accept (Poster)
**Confidence:** 5

**Metareview:**

This work proposes a novel framework for privacy preservation in the inference phase of machine learning as a service (MLaaS). The proposed method, PriFU (Privacy Filter Unit), focuses on removing task-irrelevant information from the input data without the need for adversarial learning. The framework employs gradient-based task relevance metrics, specifically Average Relative Influence (AvgRI), to measure and filter channel-level features, thereby enhancing privacy while maintaining task performance.

Strengths of this work include the introduction of the PriFU method, which captures task-relevant information without relying on adversarial learning, marking an innovation in privacy-preserving machine learning. The use of gradient-based task relevance metrics (AvgRI) to quantify and filter task-relevant features is an effective strategy to enhance privacy preservation. Extensive experiments are conducted across multiple datasets (CelebA, LFW, Cifar10) and different neural network backbones (ResNet18, ResNet152, VGG16, DenseNet18). The results consistently demonstrate the effectiveness and generality of the PriFU method. Additionally, the methodology is well-structured and clearly described, with detailed explanations of the threat model, target model, and the proposed framework. Diagrams and figures effectively support the text, enhancing understanding.